# Synthesis of Water Resistance and Moisture-Permeable Nanofiber Using Sodium Alginate–Functionalized Waterborne Polyurethane

**DOI:** 10.3390/polym12122882

**Published:** 2020-12-01

**Authors:** Wen-Chi Lu, Fu-Sheng Chuang, Manikandan Venkatesan, Chia-Jung Cho, Po-Yun Chen, Yung-Ru Tzeng, Yang-Yen Yu, Syang-Peng Rwei, Chi-Ching Kuo

**Affiliations:** 1Research and Development Center of Smart Textile Technology, Institute of Organic and Polymeric Materials, National Taipei University of Technology, Taipei 10608, Taiwan; wcl2320@mail.lit.edu.tw (W.-C.L.); manikandanchemist1093@gmail.com (M.V.); b10116006@gmail.com (P.-Y.C.); j5588942ames@gmail.com (Y.-R.T.); f10714@ntut.edu.tw (S.-P.R.); 2Department of Applied Cosmetology, Lee-Ming Institute of Technology, New Taipei City 243083, Taiwan; 3Department of Fashion and Design, Lee-Ming Institute of Technology, New Taipei City 243083, Taiwan; ymj0826@mail.lit.edu.tw; 4Department of Materials Engineering, Ming Chi University of Technology, New Taipei City 24301, Taiwan; yyyu@mail.mcut.edu.tw

**Keywords:** polyurethane, alginate, nanofibrous membrane, moisture permeability

## Abstract

The development of nontoxic and biodegradable alginate-based materials has been a continual goal in biological applications. However, their hydrophilic nature and lack of spinnability impart water instability and poor mechanical strength to the nanofiber. To overcome these limitations, sodium alginate (SA) and waterborne polyurethane (WPU) were blended and crosslinked with calcium chloride; 30 wt % of SA exhibited good compatibility. Further addition of 10 wt % calcium chloride improved the water stability to an extremely humid region. Furthermore, the stress–strain curve revealed that the initial modulus and the elongation strength of the WPU/SA and WPU/CA blends increased with SA content, and the crosslinker concentration clearly indicated the dressing material hardness resulted from this simple blend strategy. The WPU/SA_30_ electrospun nanofibrous blend contained porous membranes; it exhibited good mechanical strength with water-stable, water-absorbable (37.5 wt %), and moisture-permeable (25.1 g/m^2^–24 h) characteristics, suggesting our cost-effective material could function as an effective wound dressing material.

## 1. Introduction

Segmented polyurethanes (SPUs) are extensively used in plastic elastomers because of their excellent mechanical and chemical characteristics [1,2]. With high biocompatibility and anticoagulability [3,4,5,6,7], SPUs have been evaluated for or applied to drug delivery and to biomaterials such as cardiac pump, heart valve, intubation catheter, artificial blood vessel, dressing, and bandage [6,7,8,9,10,11]. SPUs possess a two-phase structure consisting of hard and soft segments. The hard segment reacts to diisocyanate and diol chain extenders with low molecular weight. The soft segment is generally an ester-, ether-, or nonpolar-type polyol. Because SPU monomers are various, their properties can be modified using hard- or soft-segmented contents or types of soft- or hard-segmented structure to suit the intended applications and functions [12,13,14,15,16,17].

Alginates have high biocompatibility, biodegradability, and absorption and low toxicity; thus, they have been investigated for many biomedical applications, such as wound dressing [18,19], tissue engineering [20,21], peripheral nerve regeneration [22], and drug delivery [23,24]. Alginates can be ionically crosslinked through an aqueous solution of divalent cations [21,22,23]. The network structures of covalent bonds with alginate are formed through an ion exchange reaction between the sodium carboxylates (–COONa) of alginate molecules and divalent cations in an aqueous solution such as a calcium ion aqueous solution. Several studies have indicated that the calcium alginate gel is haemostatic and mucosal protective and the gel has been used to promote wound healing and reduce blood loss from wounds [18,19]. Porous alginate dressing has been developed using an aqueous solution of sodium alginate (SA), followed by the freeze-drying of the solution under rapid freezing and subsequent immersion in a calcium chloride solution in order to allow a crosslinked reaction to form an insoluble porous gel. Alternatively, it is prepared with a mixture of an SA solution and a water-soluble polymer [25,26]; the moulding mixture of the SA/water-soluble polymer is placed in a vaporized calcium solution for gel treatment. Subsequently, the moulding mixture is dissolved in boiling water to remove the water-soluble polymer [18,19,27,28,29]. However, alginate dressing presents numerous challenges. It is hard when dry and cannot stretch, resulting in redness and damage to skin. When wet, it lacks its protective ability and may be damaged. Furthermore, the alginate dressing gel is easily disintegrated under excessive wetting, such as when the wound generates large amounts of exudate [18].

In polymer applications, one approach is polymer blending [30,31], which adopts the individual properties of the polymer to modify materials as needed. SPUs in biomedical applications have been blended with polymers such as polyvinyl alcohol [32], polyethylene glycol (PEG) [33,34], and chitin [35]. SPU/PEG-blended materials with high biocompatibility have been studied for use in small-diameter vascular tissue engineering [33,34] through electrospun technology [36,37,38,39,40,41,42,43,44]. Therefore, the SPU and the SA were blended and crosslinked, which may reduce the disintegration of alginate dressing resulting from excessive moisture absorption, skin irritation caused by SA hardness, and the lack of stretch and protection.

Ester or ether polyol has been adopted in typical thermoplastic or solvent-type soft-segment SPUs. However, these materials cannot be blended with hydrophilic SA due to their incompatibility, and they have no nontoxic common solvent to enable blending. Therefore, this study prepared a compatible waterborne polyurethane (WPU) with polycaprolactone diol (PCL) and PEG-mixed soft segment. WPUs, which are similar to typical SPUs, exhibit excellent performance in areas such as biocompatibility, elasticity, and toughness and can be used in biomedical materials [45]. The WPU and the SA were blended using water, and the WPU/SA blend dispersions were subjected to electrospinning to form a nanofibrous (NFs) membrane. The WPU/SA NFs membrane was then immersed in a CA solution to enable a crosslinked reaction to obtain an insoluble WPU/CA-blended NFs membrane. The study investigated the effects of moisture on the membrane, including water absorption and moisture permeability, as well as their compatibility and tensile properties.

## 2. Experimental

### 2.1. Materials

PCL (*M*_n_ = 2000, Sigma-Aldrich Co., St. Louis, MO, USA) and PEG (*M*_n_ = 1500, Sigma-Aldrich Co., St. Louis, MO, USA) were utilized to synthesis the soft segment part of a WPU. The hard segment of the WPU consisted of bis(cyclohexy-diisocyanate) (H_12_MDI, *M*_W_ = 262.3, Covestro Co., Leverkusen, Germany) and ethylenediamine (EA, *M*_W_ = 60, TCI Co., ToKyo, Japan). Moreover, the anionic center of and a neutralizer for WPU adopted dimethylopropionic acide (DMPA *M*_W_ = 134, TCI Co., ToKyo, Japan) and triethylamine (TEA, *M*_W_ = 115, TCI Co., ToKyo, Japan), respectively. SA (*M*_W_ = 12,000–40,000) was purchased from Da Mao Chemical Reagent Factory (Da Mao Co., Tianjin, China).

### 2.2. Synthesis of WPU

Anion-type WPU copolymers were executed by a prepolymer method. The WPU consisted of H_12_MDI, PEG, PCL, DMPA, and EA, of which the molar ratio in stoichiometry was as follow: H_12_MDI:PEG:PCL:DMPA:EA = 3:1:1:0.5:0.5.

A 3L four-neck glass flask, which was equipped with a high-speed mixer, a heater, and a quantitative pump, was employed in the synthesis of the WPU mentioned above. The H_12_MDI, PEG, and PCL in stoichiometry were initially added to a reactor at about 90 °C for 3 h to obtain a NCO-terminated prepolymer in prepolymerization, and the DMPA were added in the reactor for 1 h to form an anionic NCO-terminated WPU prepolymer. After that, the reaction was cooled down to about 45 °C, and then a small amount of an acetone solvent and stoichiometric TEA were added and stirred for 1 h for neutralization. Subsequently, the deionized water was added in the reactor by a quantitative pump to emulsify at 35 °C for 6 min, and the sample was stirred by a stirrer at a high spin speed up to 1000 rpm for 40 min to disperse uniformly. Finally, EA and H_2_O were added into the reactor for the reaction of chain extension for 2 h. Once the reaction was completed, the acetone and TEA in the WPU emulsion were removed by vacuum distillation. The process of reaction was displayed in Figure 1. The emulsification stability test of WPU copolymers was performed by centrifuge at 3000 rpm for 15 min. After the centrifugal sedimentation process was finished, the WPU showing no sedimentation indicated that the emulsion was stable for at least 6 months and the WPU emulsion was dried and made into a film for FTIR testing.

### 2.3. Preparing WPU/SA Blends and Crosslinked WPU/SA (WPU/CA) Blends

The concentration of an SA aqueous solution higher than 1.2 wt % was found difficult to be dispersed at room temperature; thus, the WPU and the SA were adjusted to 1.2 wt % aqueous solution with deionized water, respectively. Subsequently, the WPU and SA aqueous solutions were weighed and added in a breaker to be stirred for about 10 min according to Appendix A. Finally, the aqueous solutions of the WPU/SA blends were loaded into a 10 mL plastic disposable syringe. During the electrospinning process, the positive terminal of a high-voltage power source connected with a stainless steel syringe needle, and its negative terminal connected to the cupper plate collector, which was placed at 10 cm from the needle. The applied voltage, flow rate, and relative humidity were regulated to 15 kV, 1 mL/h, and 35%, respectively. The dried NFs membrane were placed in a 10 wt % CA ride aqueous solution to perform an ion exchange reaction for 20 min and then dried in an oven at 60 °C to obtain a crosslinked WPU/SA (WPU/CA) NFs membrane for further testing.

### 2.4. Measurements

#### 2.4.1. Fourier Transform Infrared Spectroscopy 

The FTIR spectra of the WPU, SA, WPU/SA NFs, and crosslinked WPU/SA (WPU/CA) NFs membrane samples were performed in a Perkin Elmer Spectrum One FT-IR spectrometer (Shimadzu, Kyoto, Japan), and the characteristic absorption of the functional group at a resolution of 4 cm^−1^ and a scanning range from 650 to 4000 cm^−1^ was collected.

#### 2.4.2. Water Solubility and Degree of Crosslinking

The sample dried at 120 °C for 20 min was weighed with an error range of 0.0001 g and placed in an ultrasonic oscillating machine filled with deionized water for 10 min. Then, the sample was taken out and dried at 120 °C for 20 min for weighing. The water solubility can be calculated as follows: water solubility (%) = (dry weight sample − dry weight after ultrasonic cleaning in the water)/dry weight sample.

The WPU/SA NFs membranes were placed in a CA solution for 20 min with different concentrations (10–50 wt %) to proceed the crosslinked reaction of the WPU/SA blend (WPU/CA). The NFs membranes of WPU/CA blends were dried at 120 °C for 20 min and weighed with an error range of 0.0001 g after washing. Subsequently, the dried WPU/CA blends were placed in an ultrasonic oscillating machine, which was filled with deionized water for 10 min at 23 ± 1 °C, and then the blends were dried at 120 °C for 20 min and weighed. The percentage of water absorption can be calculated as follows: degree of crosslinking (%) = (dry weight of (WPU/SA) sample − dry weight of (WPU/CA) sample after ultrasonic cleaning)/dry weight of (WPU/SA) sample.

#### 2.4.3. Water Absorption

All samples of NFs membranes were cut into 4 × 4 cm size and dried in an oven at 120 °C for 2 h. After cooling, the samples were weighed with an error range of 0.0001 g and placed in deionized water at 23 ± 1 °C for 2 h. Next, the samples were taken out to remove the excess water on the surface by a filter paper and weighed with an error range of 0.0001 g. The percentage of water absorption can be calculated as follows: water absorption (%) = (wet weight sample − dry weight sample)/(dry weight ample) × 100%.

#### 2.4.4. Moisture Permeability

The moisture permeability properties for all samples with a thickness of 0.25 mm were tested by a programmable temperature and humidity chamber (Pin Tai THP-S225) under 25 ± 0.5 °C and 65 ± 5% RH for 24 h. The moisture permeability (P) can be calculated as follows:P [g/(m^2^)] = (a2 − a1)/S(1)
where a2 is the weight before testing, a1 is the weight after testing, and S is the permeable area (m^2^).

#### 2.4.5. Phase Transition Regions and Degree of Phase Separation

The regions of phase transition temperature for all samples were determined by a Perkin–Elmer Pyris-1 differential scanning calorimeter (DSC). spectrometer (New Castle, DL, USA) All DSC scans were performed from −100 to 250 g/(m^2^) at a heating rate of 20 °C/min under nitrogen at a flow rate of 100 mL/min, and each sample was weighed about 5 mg. The breadth of soft-segment glass transition (ΔB = *T*_g2_ − *T*_g1_) was used as a qualitative measure of the degree of phase separation [46].

#### 2.4.6. Tensile Strength

The tensile strengths of all samples were proceed by a tensile machine (MTS-Qtest-5) at 25 °C and a relative humidity of 60 ± 5%. The morphology and elemental analyses of the WPU/SA and WPU/CA blends were investigated by a scanning electron microscope (Hitachi H-3000N CHanning Electron MicroCHope) equipped with EDS (Horiba 7021H EDS) spectrometer (Hitachi Co., Ltd., Rotkreuz, Switzerland).

## 3. Results and Discussion

### 3.1. FTIR Spectrum Analysis of the WPU/SA and WPU/CA Blends Characteristics

The urethane and urea groups of WPU hard segments were formed by reacting isocyanates with hydroxyl and amines, respectively. The urethane segments of the WPU in the study resulted from the reactions of monomers including H_12_MDI, DMPA, PEG, and PCL, and the urea segment was formed by H_12_MDI and EA. The characteristic absorptions of the urethane and urea groups were approximately 1650 cm^−1^ (carbonyl group of urea) and 1710 cm^−^^1^ (carbonyl group of urethane), respectively, as depicted in Figure 2. Notably, the characteristic absorption for the isocyanate group of H_12_MDI was observed at approximately 2350 cm^−1^, but the absorption peak of the isocyanate group clearly disappeared in the FTIR spectrum of the WPU. This result indicated that H_12_MDI nearly reacted with DMPA, PCL, PEG, and EA to form WPU. Other characteristic absorptions of groups in the WPU were observed at 3500–3200 cm^−1^ (–NH stretching vibrations), 2940–2920 cm^−1^ and 2860–2850 cm^−1^ (–CH_2_ symmetrical and asymmetrical stretching vibrations), 1350–1360 cm^−1^ and 1400–1450 cm^−1^ (–CH_2_ bending vibrations), 1150–1250 cm^−1^ (RCO–OR stretching absorptions), and 1150–1040 cm^−1^ (–C–O–C- stretching vibrations). SA is mainly composed of natural polysaccharides and is a water-soluble colloid formed by alginic acid and sodium ions. The characteristic absorption of SA included considerable numbers of hydroxyls and presented a broad absorption peak at 3000–3500 cm^−1^. Other characteristic absorption peaks were carboxylate at 1550–1610cm^−1^ (–COO–), methine at 2890 cm^−1^ (–CH stretching vibrations) and 1400 cm^−1^ (–CH bending vibrations), and ether at 1150–1040 cm^−1^ (–C–O–C– stretching vibrations).

The FTIR spectra of the WPU/SA and WPU/CA blends were similar and exhibited characteristic absorption peaks of SA and WPU, as represented in Figure 2. Notably, the absorption peak of the urethane group on the spectra of the WPU/SA and WPU/CA blends moved to the lower region at approximately 1710 cm^−1^, which indicated that the hydrogen bond is intermolecular [47,48]. Thus, WPU and SA are compatible through intermolecular hydrogen bonding. Furthermore, the absorption peak of SA carboxylate in the WPU/SA blend became quite weak at 1610 cm^−1^, which may have been due to carboxylate anions interacting with protons of the urethane or urea groups. In particular, the absorption peak of carboxylate in the WPU/CA almost disappeared, which was possibly caused by the coordination bonds resulting from ion exchange between calcium ions and SA.

### 3.2. Effect of Moisture on the WPU/CA Blends

#### 3.2.1. Solubility of the WPU/CA Blends in Water

SA has highly water solubility; thus, SA NFs membranes dissolve or swell in high-humidity states, resulting in their destruction. The WPU/SA blend NFs membranes with water solubility were placed in a 10 wt % water solution of CA to reduce water solubility. The composition of SA allowed the exchange of calcium ions. Therefore, a three-dimensional network structure can exist in SA and SA/WPU blends through ionic bonding [22,23]. Because an ion coordination bridge occurred in the SA intermolecular of the WPU/SA blends, the crosslinked WPU/SA (WPU/CA) blends, which have a semi-interpenetrating polymer network structure, may have low water solubility. As Table 1 suggests, the water solubility of the crosslinked SA (CA) had only 0.5 wt % water solubility. Moreover, the water solubility of the WPU/CA blends was less than 3.5 wt %. However, the water solubility of the WPU/SA blends was greater than 9.5 wt %, close to the weight percentage of SA in the blends; thus, the SA in the WPU was nearly dissolved in the water. This result indicated that the water solubility of the WPU/SA blends could be reduced by the crosslinking resulting from the reaction between the calcium and SA ions in the WPU/SA blend. Therefore, the structure of the WPU/CA blends could be maintained, without being destroyed after absorbing liquids. The WPU/CA_30_ in the study exhibited the lowest water solubility, which was possibly related to the morphology of the WPU/CA blends.

#### 3.2.2. Water Absorption of the Crosslinked WPU/SA Blends

SA possesses strong hydrophilic groups such as the hydroxyl group (–OH) and the sodium carboxylate group (–COONa). As mentioned, the sodium carboxylate groups (–COONa) in the WPU/SA blends were crosslinked by calcium ion exchange to form a crosslink structure, but the crosslinked WPU/SA (WPU/CA) blends retained high hydrophilicity due to the numerous hydroxyl groups. To analyze the water absorption ability of the WPU/CA blends, the dry NFs membrane of the blends were placed in deionized water at 23 ± 1 °C for 2 h. Appendix A displays the results. The crosslinked SA (CA) and WPU exhibited 139.1 and 10.2 wt % of water absorption, respectively. This result indicated that the CA remained highly hydrophilic but WPU had little hydrophilic ability as a result of the PEG soft segment. In addition, the water absorption of the WPU/CA blends increased with CA content [49], which can again be attributed to CA hydrophilia, such as that of its hydroxyl group. Notably, the relationship between water absorption and CA content for the WPU/CA blends was not linear because of the effect on the degree of crosslinking and the blend morphology, as depicted in Appendix A. For example, the WPU/CA_50_ had the highest water absorption and its absorption was more than twice that of the WPUCA_30_, because the WPU/CA_50_ had a lower degree of crosslinking than did the WPU/CA_30_ on the uniform blend morphology, which allowed generous moisture swelling.

### 3.3. Moisture Permeability of the Crosslinked WPU/CA Blends

The solubility of the water and water absorption of the WPU/SA blends were reduced by the coordination crosslink of calcium ions, but the WPU/CA blends retained water absorptions of 25.7 to 82.5 wt % due to the number of hydroxyl groups that originated from CA in the WPU/CA blends. Moisture transfer in hydrophilic materials was performed through absorption, diffusion, and dehumidification processes. Because the SA in the WPU/SA blends nearly dissolved in high humidity, the moisture permeability properties of the WPU/CA blends with different CA contents were tested for 24 h to investigate their moisture transmission abilities.

As Appendix A suggests, the moisture permeability of the WPU/CA blends increased with CA content. This result indicated that the WPU/CA blends had sufficient transport capacity, which can be attributed to the conduction of hydrophilic molecules of WPU/CA such as hydroxyl (–OH), acetyl (–NHCO–CH_3_), or a few uncrosslinked carboxyl (–COOH) or to the intermolecular crosslinked calcium alginate structure. Therefore, even if crosslinking reduced the water absorption capacity of WPU/CA blends, the moisture permeability of the WPU/CA blends was 19–34 g/m^2^ at 24 h. Furthermore, the relationship between the moisture permeability and CA content of the WPU/CA blends was not linear, as can be observed in Figure 3. For example, the moisture permeability of the WPU/CA_20_ blend was only 1 g/m^2^–24 h higher than that of the WPU/CA_10_ blend; however, with the CA content between 20 and 40 wt %, the moisture permeability of the WPU/CA increased by 5 to 6 wt % with every 10 wt % increase in CA content. Subsequently, the increased magnitude of moisture permeability, which was only 3 wt %, dropped, as the CA content of the WPU/CA blends was increased from 40 to 50 wt %. This result signified that the moisture permeability of the WPU/CA blends was affected not only by the hydrophilic molecules in the CA structure but also by the degree of crosslinking and morphological change. Increasing the number of the hydrophilic groups in the blends accelerated the rate of moisture permeability, until the blends reached saturation [50,51], which prevented absorption, diffusion, and dehumidification. Therefore, the moisture permeability of a blend may reach an optimal state when the blend attained suitable crosslinking and morphology (Appendix A). The relatively low number of crosslinked blends, such as the WPUCA_40_ and WPUCA_50_, in this study, possessed larger intermolecular pores than did the WPUCA_30_, resulting in them having high water absorption and saturated moisture absorption, which prevented the dissipation of moisture due to the strong hydrogen bond between water molecules. Therefore, when the CA content of the WPU/CA blends exceeded 40 wt %, they exhibited higher water absorption than the other blends (Appendix A), but the increased degree of moisture permeability was reduced (Figure 3).

### 3.4. Phase Transition Regions of the WPU/SA Blends and WPU/CA Blends

SPU copolymer is a microstructure with soft- and hard-segment phases. Because these dissimilar segment types are incompatible, DSC thermograms of SPU may exhibit four transition regions if both phases crystallise. Figure 4a displays the DSC thermograms of dry samples of the WPU, SA, and WPU/SA blends. Because WPU has a small hard-segment component, its DSC thermograms presented only two glass transition regions—a soft-segment glass transition temperature (*T*_gs_) region from −50 to −25 °C and a hard-segment glass transition temperature (*T*_gh_) region from 50 to 100 °C. The SA on the DSC thermogram presented a broad endothermic region at 40–150 °C, which was based on the intermolecular decomposition of numerous hydrogen bonds. The transition regions of the WPU/SA blends exhibited *T*_gs_ and *T*_gh_ regions similar to those of the WPU (Figure 4a). Notably, when the SA content was increased, both the *T*_gs_ and *T*_gh_ regions of the WPU/SA blends shifted to higher temperature regions due to the polysaccharide molecular chains of the SA in the blends, hindering WPU segment movement [52,53]. Therefore, the *T*_gs_ and *T*_gh_ regions of the WPU/SA blends increased with SA content (Table 2).

The DSC thermograms of the crosslinked SA (CA) were similar to those of SA, as can be observed in Figure 4b, but the maximum endothermic temperature of the CA (approximately 125 °C) was higher than that of SA due to the crosslinked effect of calcium ion coordination. The *T*_gs_ regions of the WPU/CA blends were broader than those of the WPU/SA blends, which demonstrated that the WPU/CA blends had higher phase separation due to the crosslink effect. The *T*_gh_ region of the WPU/CA blends was inconspicuous on the thermogram curves, such that the WPU/CA_30_ was hardly detected on them. This may result from the crosslinking of the calcium ion coordination possibly passing through or connecting with SA molecules and hard-segmented WPU domains, resulting in the crosslink inhibiting the free rotation of the hard segment. The thermograms of the WPU/CA blends present a wide continuous heat absorption area from −50 to 200 °C, indicative of secondary forces at the intermolecular level, including Van der Waals and the continuous disintegration of hydrogen and ionic bonds during heating.

The degree of phase separation affected WPU mechanical strength, because the hard-segment domains physically crosslinked and reinforced the filler particles of a soft, rubbery segment matrix. As mentioned, the WPU/CA blends had lower degrees of phase separation than did the WPU/SA. As Table 2 suggests, the *T*_gs_ and the *T*_gh_ for the WPU/SA and WPU/CA blends increased with SA or CA content, because the SA or the crosslinking of the calcium ion coordination inhibited the motion of WPU molecular chains. A high Tg temperature indicated that the segments had low mobility but did not indicate the degree of phase separation. ΔB calculated by (*T*_g2_ − *T*_g1_) was used as a quantitative measure of the degree of phase separation [46]; increased ΔB indicated that the degree of phase separation decreased. As Table 2 suggests, the WPU/CA_30_ blend in this study had the maximum ΔB value, indicating that it had the lowest degree of phase separation. However, it had a high degree of segment mixing and good compatibility along the segments of the macromolecular chains.

### 3.5. Tensile Strength

The WPU elastomer generally presents high tensile strength, because its hard-segment phase provides tensile strength and elasticity whereas the soft-segment phase allows a high degree of extensibility. SA consisting of β-D-mannuronic acid and α-L-guluronic acid is a polysaccharide, and the cyclic structure and intermolecular attraction restrict the movement of the segments at the macromolecular level. SA, therefore, presents characteristics of hard materials, such as stiffness and less elongation, and can cause skin redness or swelling when dry. A clear difference exists between rigid and soft materials in the stress–strain curve. As Figure 5a indicates, the stress–strain curve of the WPU was near its strain axis, with a high elongation and a low initial modulus. By contrast, the SA and the CA were close to their stress axes, which indicated that they had a high initial modulus and a little elongation. As Table 3 indicates, the WPU had a high elongation rate of 529% and a low initial modulus of 6.9 MPa, whereas the SA had a high initial modulus of 797.2 MPa and a low elongation rate of 15.8%. The CA had a higher initial modulus and a lower elongation rate than did the SA, because the crosslinking further limited the movable length of macromolecular chains consisting of β-D-mannuronic acid and α-L-guluronic acid.

The initial moduli of the WPU/SA and WPU/CA blends increased with SA or CA content (Table 3), which illustrated that the hardness of the blends increased as SA content increased. Furthermore, the initial moduli of these blends were lower than those of CA or SA, except for the WPU/CA_50_, which, due to its high crosslinking, resulted in the sample having a higher initial modulus than did the SA. The results indicated that the blends with SA of less than 40 wt % presented greater softness than did the SA or CA. The variation of strength is generally opposite to that of elongation with regard to WPU tensility; as the WPU had a hard composition, its elongation was low, but its strength was high. As Figure 5b indicates, the elongation of the WPU/SA blends was nearly higher than that of the SA; however, elongation plummeted in blends with SA content above 30 wt %. Similarly, the WPU/CA blends, especially the WPU/CA_40_ and WPU/CA_50_ blends, presented less elongation. The strength and elongation at break for the blends that had more than 40 wt % SA content presented a large decrease or increase due to the strong effect of blending morphology, in addition to the effect of blending composition and crosslinking. The tensile strength of the blends in this study was affected by blending composition, blending pattern, and crosslinking. Accordingly, no obvious correlation existed between the tensile strength and the SA content of the blends. However, the blends’ tensile strength had an optimal value at a 30 wt % SA content (WPU/SA_30_ and WPU/CA_30_ blends), as indicated in Table 3. The WPU/CA blend with an SA content of higher than 40 wt % presented little elongation owing to a highly crosslinked effect and the blending morphology. The crosslinking effect on the tensile strength of a blend resulted from the calcium ion coordination bonds in the WPU domains forming a three-dimensional network crosslinked structure. However, the crosslinking effect also prevented the SA from dissolving in a high-moisture environment. The increase in the density of the network structure reduced the length of the crosslinked portion between the chain units, resulting in eliminating flow and transfer along the molecular chains. Accordingly, WPU and SA blended and crosslinked with CaCl_2_ and resulted in the WPU losing its original high elasticity and elongation. Thus, the WPU/CA blends exhibited higher initial moduli and less elongation rates than did the WPU.

WPU/SA_30_ and WPU/CA_30_ presented appropriate tensile strengths, which was also related to the compatibility and morphology of the blends. (The morphology of the blends is discussed in the next section.) The compatibility and the degree of phase separation were inversely proportional. As mentioned, the ΔB values for WPU/SA_30_ and WPU/CA_30_ before and after crosslinking, respectively, presented the maximum values, which indicated that both blends had a low degree of phase separation. However, they had greater compatibility than that of the other blends. High compatibility is beneficial for the mechanical properties of a blend. Accordingly, a high degree of compatibility enabled the crosslinking of calcium ions to uniformly change the tensile strength of WPU. However, the high degree of macrophase separation of the blends reduced their tensile strengths due to the aggregate regions of another phase becoming stress concentration points. High compatibility increased the tensile strength of the blends, which is a different effect than increased microphase separation of SPU improving mechanical strength. The increase in the microphase separation of SPU improves tensile strength, based on the physical crosslinking of the hard-segment domains and the reinforcement of the filler particles in the rubbery soft-segment matrix.

### 3.6. SEM and EDS Analysis of Blends

SEM with EDS was used to analyse the surface morphology of the blends. The morphology and EDS of the cross-sections of the WPU/SA and WPU/CA blends were broken under the quenching conditions of liquid nitrogen. As Figure 6 and Appendix A reveal, the WPU/SA blends featured smoother cross-sections than did the WPU/CA blends. Moreover, the cross-sections of the WPU/SA and WPU/CA blends had more obvious rough texture or aggregated block, as SA or CA content increased. Indeed, the morphologies of the aggregated block and rough texture may result in decreased strength and elongation. The type of aggregated block and rough texture was related to blending compatibility and degree of crosslinking, respectively. A large aggregated block caused decreases in the tensile strength of the blends. A high degree of crosslinking increased a blend’s initial modulus but decreased its elongation and strength at break. The tensile strengths of the WPU/CA and WPU/SA blends were lower than that of the raw WPU due to the molecular structure of hard SA and crosslinking. The effect of crosslinking also resulted in the elongation of the WPU/CA being lower than that of the WPU/SA with the same blended content, except for the blending compatibility. As Table 3 indicates, WPU/CA_40_ had the lowest tensile strength due to the crosslinking effect and a large aggregated spherical block. Appendix A displays the SEM morphology of the WPU/CA_40_ blend. It had a rough texture and aggregated spherical block, which may have been caused by the crosslinking of ion coordinate bonds and a low degree of compatibility that caused it to have the lowest tensile strength. The WPU/CA_30_ blend featured some rough texture, but it had no excessive aggregation (Appendix A). Therefore, the WPU/CA_30_ blend possessed the highest strength and elongation at the break of the WPU/CA blends (Table 3), due to its more suitable content of the crosslinked SA and a higher degree of compatibility than the other WPU/CA blends.

The ion exchange between SA and calcium could be further presented through EDS. Appendix A depicts the characteristic peak of Na on the EDS spectrum of the WPU/SA blends. The EDS spectra of the WPU/CA blends, in which the ion exchange between sodium and calcium in alginate is complete, had only a Ca signal peak and no Na signal. This result indicated that the WPU/CA blends underwent a calcium ion coordination crosslinking through the process of ion exchange between SA and calcium.

### 3.7. Electrospinning of the WPU/SA Blends

To address the porous alginate dressing disintegrating under excessive wetting and lack of stretch and protection with mechanical damage, the alginate was blended with WPU and crosslinked with CA. The water solubility and tensile strength properties of the WPU/SA and WPU/CA blends were investigated as described. The WPU/CA_30_ blend had an optimal performance; thus, it was used to execute electrospinning and crosslinking to prepare insoluble porous gel alginate dressing. An NFs membrane of the WPU/SA_30_ blend with an SA content of 30 wt % was spun under a spinning voltage of 15 kV, and then the crosslinking was performed using an atomized CA aqueous solution. As Figure 7 displays, the features of the fibrous membrane presented random overlapping, with pores with sizes of less than 10 μm and a fiber diameter range of 350–1.07 um. Although the uniformity of fibrous diameters was not yet achieved, a porous membrane consisting of an NFs membrane of the WPU/SA blend was created through the electrospinning process. The use of electrospinning to make a more uniform WPU/CA_30_ blend porous dressing and its properties will be further discussed in future work to evaluate its application for wound dressing.

## 4. Conclusions

A water-insoluble porous nanofibers membrane was produced by blending SA with waterborne WPU synthesized through the electrospinning method followed by the effective crosslinking of CA. The resulting WPU/CA blends were almost insoluble (water stable) and retained absorbability and moisture permeability. Furthermore, the absorbability and the moisture permeability of the WPU/CA blends increased with CA content. The intermolecular crosslinking of the WPU/SA blends reduced the elongation of the WPU/CA blends but increased its initial modulus. The initial modulus increment with an increase in SA content indicated the effective reduction in the softness of blends. Our study suggestd that the tensile strength, absorbability, and moisture permeability of the WPU/CA blends were related not only to SA content and degree of crosslinking but also to blending compatibility. The WPU/CA_30_ blend had an optimal tensile strength of 8.2 MPa at break and an elongation rate of 11.9% at break due to its good compatibility. However, the WPU/CA_30_ blend still exhibited water absorption (37.5 wt %) and moisture permeability (25.1 g/m^2^–24 h). Our novel WPU/CA_30_ blend has the potential to overcome the drawbacks of SA dressing materials, including lack of extensibility and excessive water solubility under high moisture which result in loss of protective ability. In the future, this work represents an imperative stride in fabricating an effective wound dressing material based on our cost-effective material.

## Figures and Tables

**Figure 1 polymers-12-02882-f001:**
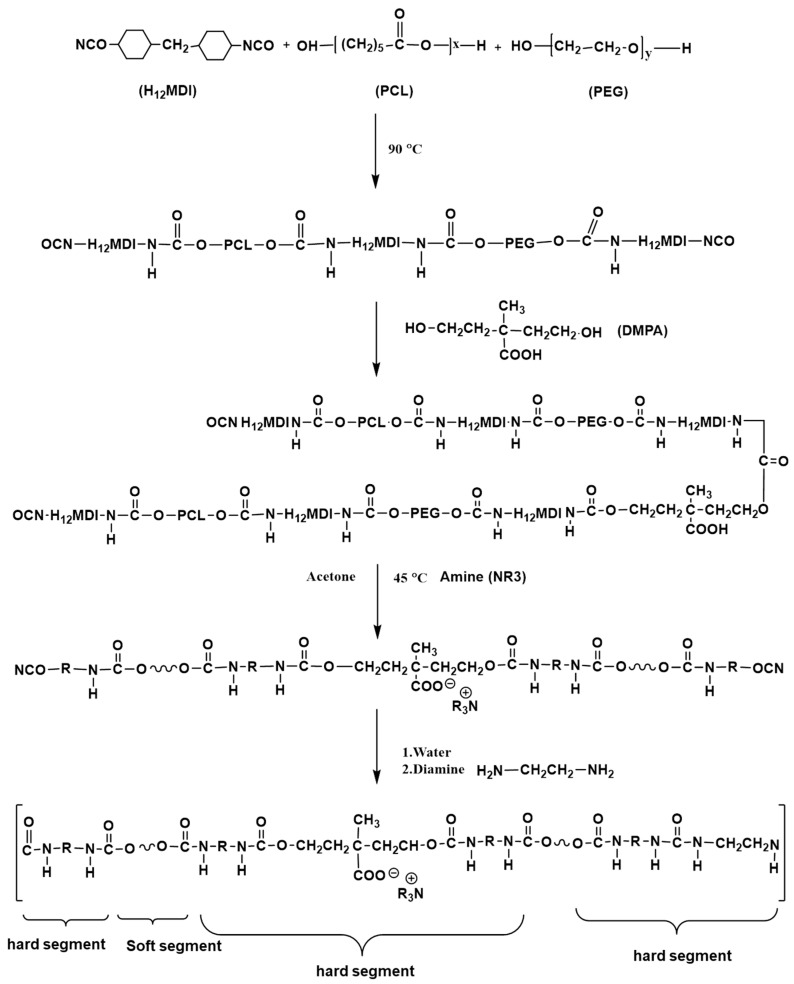
The synthesis process of a waterborne polyurethane (WPU).

**Figure 2 polymers-12-02882-f002:**
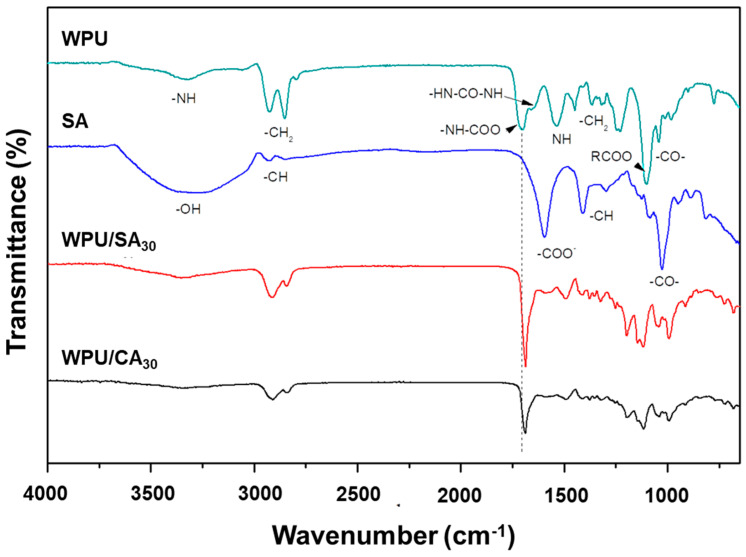
FTIR spectra of the WPU, WPU/ sodium alginate (SA), and WPU/calcium chloride (CA) blends.

**Figure 3 polymers-12-02882-f003:**
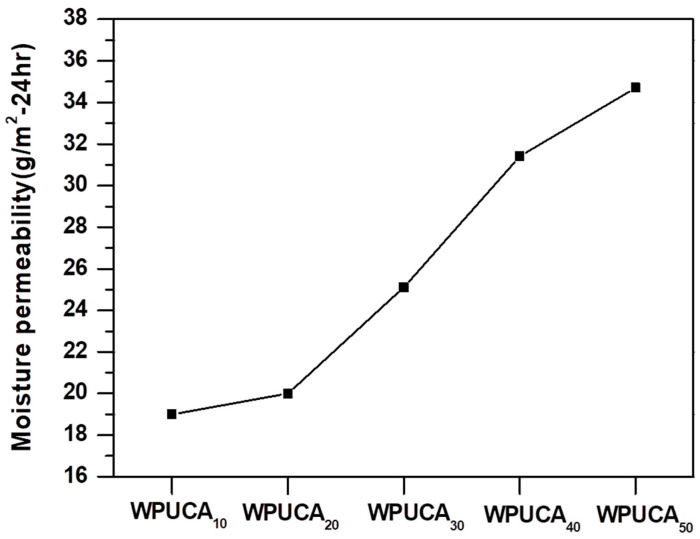
Moisture permeability of the blends with different WPU/CA weight ratios.

**Figure 4 polymers-12-02882-f004:**
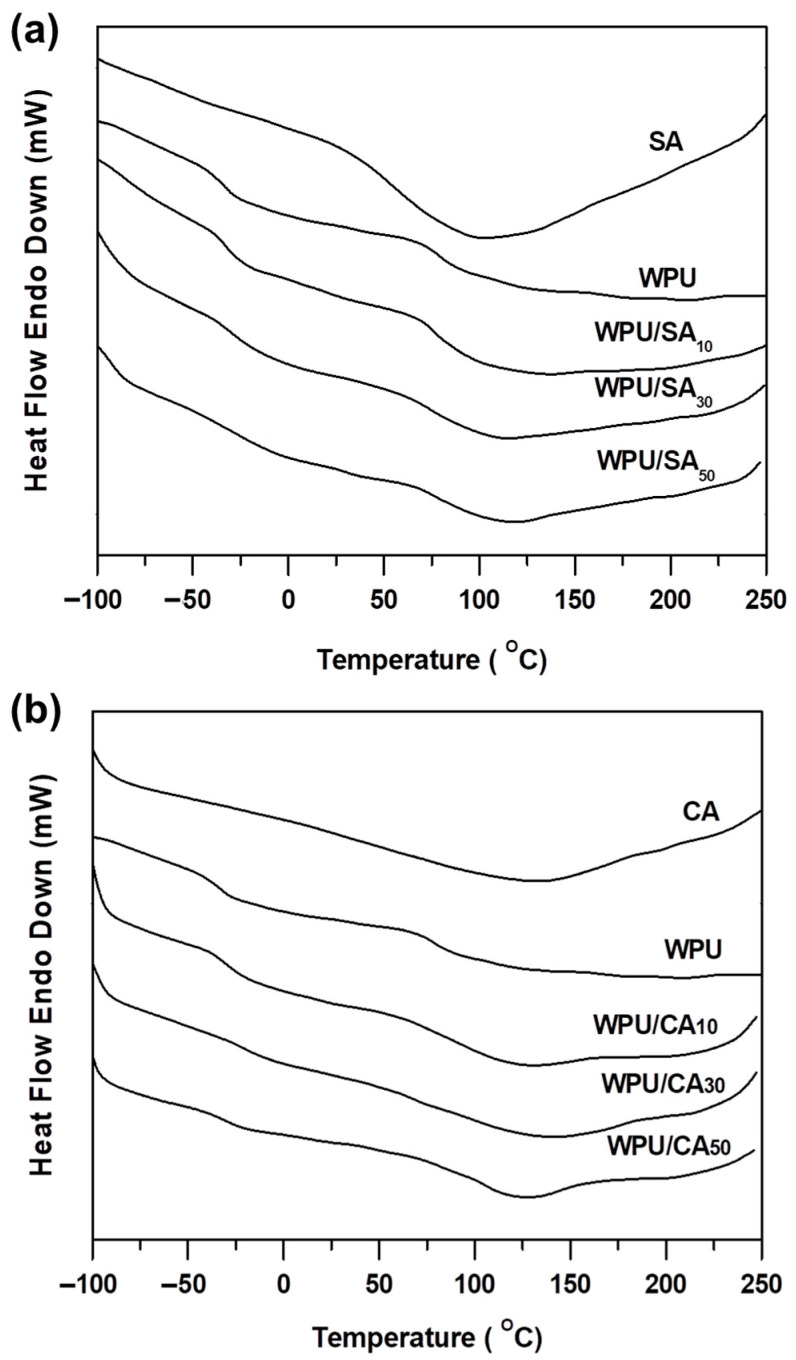
Differential scanning calorimeter (DSC) thermograms of the WPU, SA, and WPU/SA blends (**a**) and the WPU, CA, and WPU/CA blends (**b**).

**Figure 5 polymers-12-02882-f005:**
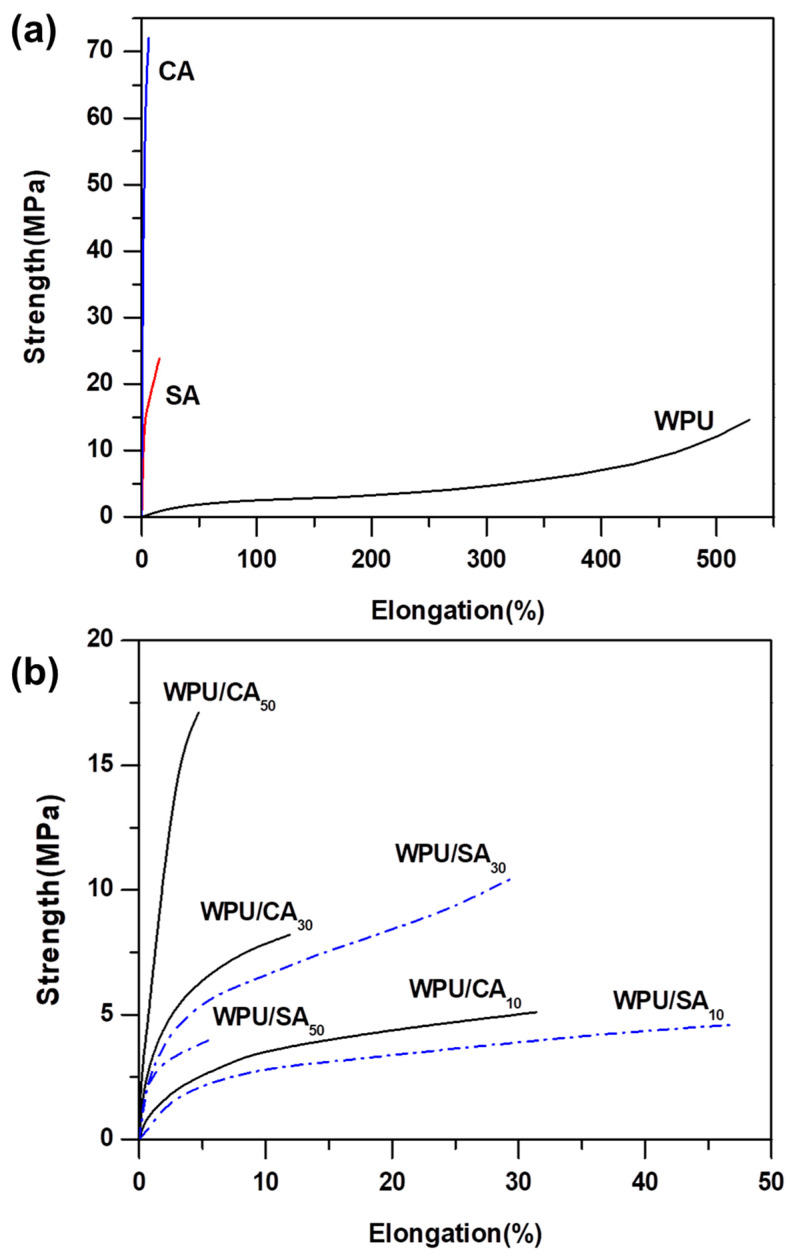
Tensile strength curves of the CA, SA, and WPU blend (**a**) and the WPU/SA and WPU/CA blends (**b**).

**Figure 6 polymers-12-02882-f006:**
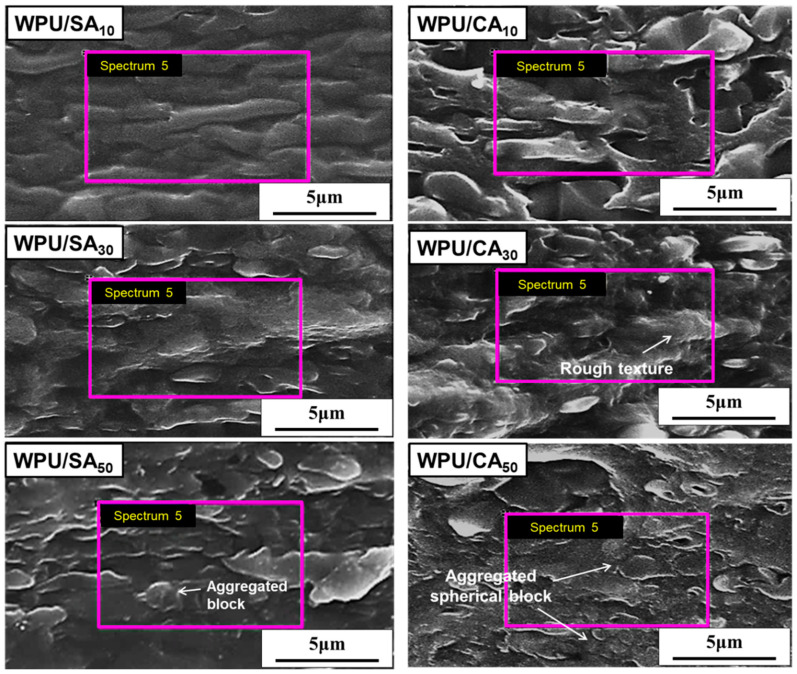
SEM morphology of the cross-sections of the WPU/SA blends and WPU/CA blends.

**Figure 7 polymers-12-02882-f007:**
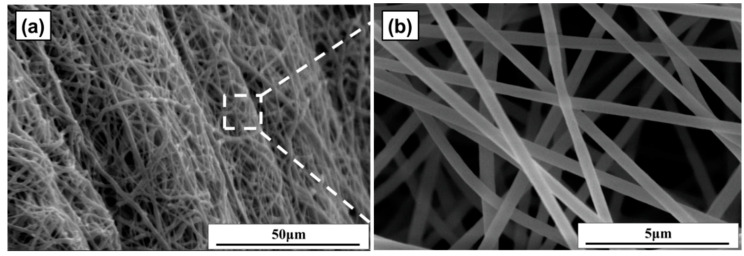
SEM images of the fiber membranes WPU/SA_30_ blend view (**a**); the enlarged view of these fibers (**b**). The scales bars in (**a**,**b**) are 50 and 5 µm, respectively.

**Table 1 polymers-12-02882-t001:** Water solubility of the WPU/CA blends.

Code	Water Solubility (wt%)	Exchange Degree of Sodium Alginate and Calcium Ions (Degree of Crosslinking)
SA ^1^	dissolve	-
WPU/SA_10_	9.5	-
WPU/SA_20_	19.8	-
WPU/SA_30_	28.6	-
WPU/SA_40_	39.4	-
WPU/SA_50_		-
CA ^1^	0.5	99.5
WPU/CA_10_	2.2	97.8
WPU/CA_20_	3.4	96.6
WPU/CA_30_	1.01	99.0
WPU/CA_40_	1.68	98.3
WPU/CA_50_	1.6	98.4

^1^ SA: sodium alginate; CA: crosslinked SA; percentage of dissolved weight in water (%) = dry weight of the sample before crosslinking − dry weight of the crosslinking sample after ultrasonic cleaning in the water/Dry weight of sample before crosslinking.

**Table 2 polymers-12-02882-t002:** Phase transition temperature (°C).

Code	*T* _gs_	*T* _gh_	ΔB
WPU	−45	71	25
SA	-	-	-
WPU/SA_10_	−39	80	33
WPU/SA_20_	−33	82	37
WPU/SA_30_	−30	87	51
WPU/SA_40_	−27	90	42
WPU/SA_50_	−25	91	50
CA	-	-	-
WPU/CA_10_	−26	97	36
WPU/CA_20_	−21	100	42
WPU/CA_30_	−11	-	52
WPU/CA_40_	−15	103	44
WPU/CA_50_	−18	105	42

*T*_gs_: soft-segment glass transition temperature; *T*_gh_: hard-segment glass transition temperature; the equation of △B = *T*_g_ − *T*_g_ was used to determine *T*_g_ and *T*_g_.

**Table 3 polymers-12-02882-t003:** Tensile strengths of the WPU, SA, CA, WPU/SA, and WPU/CA blends.

Code	Initial Modulus (MPa)	Break Strength (MPa)	Elongation at Break (%)
WPU	6.9	14.6	529
SA	797.2	23.8	7.8
CA	2516.2	72.4	4.9
WPU/SA_10_	112.1	4.6	47.1
WPU/SA_20_	154.0	5.9	21.9
WPU/SA_30_	166.6	10.6	30.1
WPU/SA_40_	196.0	5.0	16.3
WPU/SA_50_	243.3	4.1	6.1
WPU/CA_10_	98.6	5.1	31.4
WPU/CA_20_	340.6	7.7	9.4
WPU/CA_30_	403.1	8.2	11.9
WPU/CA_40_	539.4	3.1	4.3
WPU/CA_50_	1378.1	17.1	4.7

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
