# Peer review of "Synthesis of Water Resistance and Moisture-Permeable Nanofiber Using Sodium Alginate–Functionalized Waterborne Polyurethane"

_polymers, 2020, doi:10.3390/polym12122882_

Round 1
Reviewer 1 Report
This manuscript reports the effects of the water stability, absorption, and moisture permeation of the sodium alginate (SA)-functionalized waterborne polyurethane (WPU) nanofibers with calcium chloride crosslinking on their water compatibility and mechanical strength. The authors adopted a step by step method of preparing WPU first, followed by WPU/SA and WPU/SA/CA. Subsequently, the authors performed testing methods to measure the water solubility, cross-linking degree, water absorption, moisture permeability, phase separation and tensile strength. The results and discussion were explained in detail with required images and results. I would recommend the acceptance of this manuscript after a minor revision as mentioned below:
- Can the authors clarify the difference between SPU and WPU? Both are used in the introduction section. Why are the authors using WPU and not SPU? Also, it is better if the author gives the background of WPU which they have used in this work.
- Line 92 – “method” spell correction
- How did the authors arrive at 10 wt% calcium solution for crosslinking?
- Could the authors give references for the sections 3.2, 3.3 and 3.4?
- In section 2.3, the authors mentioned using only 10 wt% calcium chloride solution to obtain crosslinked samples. But the results have samples with 10, 20, 30, 40 and 50 wt% calcium chloride crosslinked i.e. WPU/CA10, WPU/CA20, WPU/CA30, WPU/CA40 and WPU/CA Please explain.
- Spelling corrections and English check is to be done
Author Response
A point-by-point revision according to the reviewers' comments
Reviewer 1
This manuscript reports the effects of the water stability, absorption, and moisture permeation of the sodium alginate (SA)-functionalized waterborne polyurethane (WPU) nanofibers with calcium chloride crosslinking on their water compatibility and mechanical strength. The authors adopted a step by step method of preparing WPU first, followed by WPU/SA and WPU/SA/CA. Subsequently, the authors performed testing methods to measure the water solubility, cross-linking degree, water absorption, moisture permeability, phase separation and tensile strength. The results and discussion were explained in detail with required images and results. I would recommend the acceptance of this manuscript after a minor revision as mentioned below:
- Can the authors clarify the difference between SPU and WPU? Both are used in the introduction section. Why are the authors using WPU and not SPU? Also, it is better if the author gives the background of WPU which they have used in this work.
Answer: Since you asked about the differences between SPU and WPU we have made some corrections in the introduction.
However, these materials cannot be blended with hydrophilic SA due to their incompatibility, and they have no nontoxic common solvent to enable blending. Therefore, this study prepared a compatible waterborne polyurethane (WPU) with polycaprolactone diol (PCL) and PEG-mixed soft segment. Waterborne polyurethanes, which are similar to typical SPUs, exhibit excellent performance in areas such as biocompatibility, elasticity, and toughness and can be used in biomedical materials [45]
Waterborne polyurethanes (WPU) and segmented polyurethanes (SPUs) are the same polyurethane derivatives that are similar to exhibit excellent biocompatibility, elasticity, and toughness and can be used in biomedical materials. However, the SPU has incompatible nature with hydrophilic solvents. Therefore, SPU cannot be used to blend with hydrophilic SA. (ref. 45) page 4
- Line 92 – “method” spell correction
Answer: Thanks for the reviewer's deep consideration of our manuscript. We corrected the spelling errors in our revised manuscript.
- How did the authors arrive at 10 wt% calcium solution for crosslinking?
Answer: Thanks for the reviewer comment. The ultimate aim of this work is to improve the moister permeability and water absorption ability with the help of a calcium precursor crosslinker. However, too much calcium and long period crosslinking make the nanofiber rigid, it strongly affects the elongation strength (refer table 3). Therefore, it cannot be used for practical applications. 10 wt% at 20 minutes is the optimized ratio. It shows the improved elongation strength (31.4), which is higher than the others and also has reasonable water absorption and moister permeation values.
- Could the authors give references for the sections 3.2, 3.3 and 3.4?
Answer: Thanks for the reviewer’s suggestion. We considered and cited the recent relevant journal papers as page 11 (ref. 49), page 12 (ref.50, 51), and page 13 (ref.52, 53).
- In section 2.3, the authors mentioned using only 10 wt% calcium chloride solution to obtain crosslinked samples. But the results have samples with 10, 20, 30, 40 and 50 wt% calcium chloride crosslinked i.e. WPU/CA10, WPU/CA20, WPU/CA30, WPU/CA40and WPU/CA Please explain.
Answer: Thanks for the reviewer comment. As prepared nanofiber exhibited water instable. Therefore, to overcome this we preferred the calcium crosslinking reaction; it also improves the water absorption and moister permeability. To confirm this, we carried out various concentrations of calcium precursor from 10-50 wt% such as WPU/CA10, WPU/CA20, WPU/CA30, WPU/CA40 and WPU/CA. Among them, 10 wt% (WPU/CA10) crosslinking had a better ability for application purposes (refer comment 3).
We also correct our manuscript in section 2.3.
The WPU/SA NFs were placed in calcium chloride solution for 20 minutes with different concentration (10-50 wt%) to proceed the cross-linked reaction of WPU/SA blend (WPU/CA).
- Spelling corrections and English check is to be done
Answer: Thanks for the reviewer comment. We made several sequence of English corrections internally and externally with the research based academic writers followed by Wallace English correction service. Thereby we assure that revised version will serve better for the readers and reach the audience attention.

Reviewer 2 Report
Synthesis of water resistance and moisture permeable nanofiber using sodium alginate functionalized waterborne polyurethane
Manuscript presents good research work related sodium alginate functionalized waterborne polyurethane. It is recommended for publication after following minor changes.
- Abstract should contain some quantitative information also.
- English must be improved.
- Novelty of the work be established.
- All the results reported be compared in a tabular form to establish the superiority of the work.
- Authors must need to incorporate some recent references related to nanofibers in the introduction part of the manuscript related to the theme to make it more interesting for the readers. For example;
- Rev. 2018, 118, 24, 11575–11625
- ACS Sustainable Chemistry & Engineering 6 (3), 3279-3290
- Industrial & Engineering Chemistry Research 56 (46), 13885-13893
- Biomacromolecules 18 (8), 2333-2342
- https://doi.org/10.3389/fchem.2020.00392
- Cellulosevolume 26, pages7585–7617(2019)
- Soc. Rev., 2018,47, 2837-2872
- ACS Sustainable Chem. Eng.2019, 7, 19, 15800–15827
- DOI: 10.1016/j.mattod.2018.02.001
- Advanced Sustainable Systems, 1900114
- Authors need to include future prospective of the work in the conclusion part of manuscript.
Author Response
Reviewer 2
Synthesis of water resistance and moisture permeable nanofiber using sodium alginate functionalized waterborne polyurethane
Manuscript presents good research work related sodium alginate functionalized waterborne polyurethane. It is recommended for publication after following minor changes.
- Abstract should contain some quantitative information also,
Answer: Thanks for the reviewer's suggestion. The aim of this work is to prepare the water-resistive nanofibrous membrane consisting of the water absorption and moister permeation ability. These values are the most basic requirements for the wound healable materials synthesis. However, too many numerical values will divert the readers' attention. Moreover, abstract word counts rise unnecessarily. Considering all of this, we provide quantitative information only for water absorption and moister permeations. Apart from this the quantitative information are well organized in tables. Therefore, this abstract serves the best demonstrating our concepts and this will definitely attract readers from the diversified fields.
- English must be improved.
Answer: Thanks for the reviewer comment. We made several sequence of English corrections internally and externally with the research based academic writers followed by Wallace English correction service. Thereby we assure that revised version will serve better for the readers and reach the audience attention.
- Novelty of the work be established.
Answer: Thanks for the reviewer comment. The novelty of this work is to prepare a water resistance electrospun nanofiber membrane by using various concentrations of calcium chloride crosslinking agent. The prepared membrane was subjected to water absorption and moisture permeation test to prove its ability towards application. An extensive study was accomplished with mechanical and thermal behaviors.
- All the results reported be compared in a tabular form to establish the superiority of the work.
Answer: Thanks for the reviewer comments. We are pleased to receive the encouraging positive comments.
- Authors must need to incorporate some recent references related to nanofibers in the introduction part of the manuscript related to the theme to make it more interesting for the readers.
Answer: Thanks for the reviewer's suggestion. The reviewer suggested articles were considered. We made better connections with the suggested recent articles demonstrating the state of art results and breakthroughs thereby improved our manuscript quality to a good extent. Page no: page 3 (ref.11, 24), page 4 (ref.28, 29, 30, 31, 41, 42, 43, 44)
- Authors need to include future prospective of the work in the conclusion part of manuscript.
Answer: Thanks for the reviewer's suggestion. We made the future prospects of the work in the conclusion part of the manuscript.
A water-insoluble porous nanofibers membrane was produced by blending SA with waterborne WPU synthesized through the electrospinning method followed by effective crosslinking of calcium chloride (CA). The resulting WPU/CA blends are almost insoluble (water stable) and retain absorbability and moisture permeability. Furthermore, the absorbability and moisture permeability of the WPU/CA blends increases with CA content. Intermolecular crosslinking of the WPU/SA blends reduces the elongation of the WPU/CA blends but increases its initial modulus. The initial modulus increment with an increase in SA content indicates the effective reduction in softness of blends. Our study suggests that the tensile strength, absorbability, and moisture permeability of the WPU/CA blends are related not only to SA content and degree of crosslinking but also to blending compatibility. The WPU/CA30 blend has the optimal tensile strength of 8.2 MPa at break and elongation of 11.9% at break due to its good compatibility. However, the WPU/CA30 blend still exhibits water absorption (37.5 wt%) and moisture permeability (25.1 g/m2-24 h). Our novel WPU/CA30 blend promises to overcome the drawbacks of SA dressing material, including lack of extensibility and excessive water solubility under high moisture resulting in loss of protective ability. In the future, this work represents an imperative stride in an effective wound dressing material based on our cost-effective material by opening up new possibilities.
